# OpenReview forum: "U-RankMOEA: Learning to Optimize High-Dimensional Expensive Multi-Objective Problems"
_ICLR.cc/2026/Conference — ICLR 2026 Conference Withdrawn Submission_

### Official Review · Reviewer_wj5j · 2025-10-21

**Soundness:** 2
**Presentation:** 2
**Contribution:** 2
**Rating:** 2
**Confidence:** 3

**Summary:**

The paper proposes a framework of multi-objective optimization called U-RankMOEA, which is a hybrid type approach of the Bayesian modeling and evolutionary strategies. U-RankMOEA combines a variety of components such as rank-based classifier of nondominant sorting, complexity reduced deep GP, and a fitted acquisition function.

**Strengths:**

The paper constructs a practical multi-objective optimization algorithm by combining a variety of component techniques, such as rank-based classifier, evolutionary procedure, deep GP, and fitting of historical acquisition function values.

**Weaknesses:**

The proposed method is a collection of approximation strategies, and overall, each of components does not have particular technical novelty.

In my understanding, in the end, the proposed method is an approximate EHVI. In this sense, recent EHVI based methods should have been in baselines. Gradient-based optimization of EHVI has been studied and therefore many (time consuming) expected volume computations may not be required. Further, in the case of the bi-objective case (for which many results are provided), the volume computation is not so difficult compared with higher output dimension case.

For the rank-based model, the model itself is directly from an existing method, and so, a contribution may be in uncertainty quantification, but a rationale behind the uncertainty quantification is not clear.

Although a theoretical analysis is shown in appendix, it is quite sudden (nothing is explained in the main text). The technical descriptions in the analysis are too messy by which I couldn't follow the detail (in my current understanding, the complete proof is not provided). Currently, I don't think the analysis is reliable.

In H.2, the assumption (A5) seems too strong (though as mentioned above, I couldn't fully follow the proof).

Minor:
- 'K nondomination rank categories' is not defined. Predicting this nondomination rank is probably directly from an existing study, but to be self-contained, more detailed definition should have been provided.
- About the theoretical analysis, even for the exact EHVI, the theoretical guarantee has not been widely studied (I only know the case of a scalarized acquisition). Since no relationship with existing analysis is revealed for the analysis, I do not find how the analysis is interpreted in the context of related theoretical analysis.
- The proposed method selects a next point by an approximate EHVI and evolutionary strategies are in the acquisition function optimization. Then, it seems the proposed method should be called Bayesian optimization rather than an evolutionary algorithm (EA). I thinks the name U-RankMO'EA' is confusing.

**Questions:**

I don't understand why mutual information (7) can be seen as an epistemic uncertainty. Further, (7) is not mutual information.

The first term of (7) is entropy of \bar{p}, and the second term is the average entropy of p^s for each s \in [S(x)]. Therefore, both of them is seemingly not the uncertainty derived by MC dropout (e.g., it should reflect the variation of p^s over different s). This makes interpretation of u_ep^clf unclear and unreliable.

I don't understand technical novelty of the uncertainty decomposition of complexity-reduced deep GP. What is the technical difficulty to decompose the uncertainty in deep GP?

How to estimate \hat{s}_div is not written.

How is K selected?

In Table 4 (ablation study), the proposed U-RankMOEA shows best performance. However, some components in U-RankMOEA are introduced for reducing computational complexity, not for improving sample-efficiency (iteration-efficiency) in my understanding. Why the results are improved? (I assume the authors employed the 300 max evaluations setting described in Sec4.1).

In SecH.2, what does 'surrogate+acquisition fidelity' in (A5) mean?

The analysis in SecH seemingly does not depend on rank-conditioned offspring generation. Why?

---

> ### Author Response · Authors · 2025-11-19
> **We sincerely hope to receive your support and encouragement!**
>
> # Response to Reviewer wj5j
>
> Dear Reviewer wj5j, thank you for the careful and constructive review. Your comments helped us identify places where the presentation can be improved and where additional experiments would strengthen the manuscript. Below we respond to each of your substantive points, clarify technical issues, and describe the exact revisions and experiments we will add. **We hope this rebuttal helps resolve the concerns raised by the reviewers. We sincerely hope to receive an improvement in your score**.
>
> ---
>
> # 1.High-Level Response (Summary)
>
> U-RankMOEA is not merely a collection of known approximations; its contribution is a theory-guided co-design that:
>
> - **Uses calibrated classifier uncertainty** to cheaply propose rank-conditioned candidates.
> - **Employs expressive yet tractable Deep Gaussian Process surrogates** that return a decomposed uncertainty signal.
> - **Introduces a lightweight history-aware acquisition learner** to correct systematic surrogate biases.
>
> All components are orchestrated to maximize hypervolume under extremely tight evaluation budgets. The novelty lies in how these modules are designed together and how uncertainty is managed end-to-end to improve iteration efficiency in very high-dimensional decision spaces.
>
> Below we respond to your detailed points.
> # 2.“No particular technical novelty; this is an approximate EHVI” , response
> ## Clarification on Novelty
>
> We agree that some building blocks such as MC-Dropout, sparse Gaussian Processes, random Fourier features, and evolutionary search are individually known. The novelty lies in three tightly coupled aspects:
>
> - **Finite-budget hypervolume decomposition**: Appendix H introduces a decomposition that identifies separable error terms: classifier error, surrogate error, and acquisition error, and shows how each term contributes to expected hypervolume shortfall. This decomposition serves as the design principle that explains why the three modules are necessary and complementary.
>
> - **Classifier-first screening and generation pipeline**: This pipeline uses calibrated epistemic signals to cheaply explore extremely large decision spaces (millions of candidates) before any costly surrogate evaluation. Without this front end, expressive surrogates cannot be used at scale under the 300-evaluation budget.
>
> - **Practical Deep Gaussian Process implementation**: The implementation preserves epistemic and aleatoric decomposition while scaling via sparse variational inference, adaptive inducing points, low-rank outputs, and selective random feature substitutions (RFF and Nyström). Maintaining a principled variance decomposition in deep hierarchies is nontrivial, as detailed in Section 3.4 and Appendix H, and is essential to feed correct uncertainty to the acquisition learner.
>
> While individual techniques are known, their principled integration combined with the finite-budget analysis constitutes a technical contribution that meaningfully advances sample-efficient high-dimensional expensive multi-objective optimization.
> # 3. “Why Deep GPs? What is difficult about decomposing uncertainty in deep GPs?”
>
> ## Clarification on Deep Gaussian Processes
>
> **Short answer**: Deep Gaussian Processes are required to obtain both expressive learned representations of high-dimensional structure and principled epistemic versus aleatoric uncertainty. The difficulty is that exact inference in deep Gaussian Processes is intractable, and naïve approximations can entangle epistemic and aleatoric terms.
>
> Concretely, for any probabilistic surrogate we want the predictive variance decomposed as:
>
>
> **Var[y | x] = Eq[Var[y | f]] (aleatoric) + Varq[E[y | f]] (epistemic)**
>
>
> where \( q \) denotes the variational posterior over latent functions \( f \). In a deep Gaussian Process the latent \( f \) is hierarchical and \( q \) is approximated with structured variational factors. Estimating both terms accurately requires careful Monte Carlo over the variational posterior and bookkeeping of layerwise contributions.
>
> **Our contribution** is a scalable estimator that:
> - Uses a limited number of variational samples and rank-reduced output covariances to estimate both terms with controlled bias.
> - Uses adaptive inducing points and selective random Fourier feature layers to keep the estimator cost practical while preserving the decomposition used by the acquisition learner.
>
> These details are explained in Section 3.4 and formalized in Appendix H. We will expand Section 3.4 in the revision to make the technical steps more explicit.

---

> ### Author Response · Authors · 2025-11-19
> **We sincerely hope to receive your support and encouragement！**
>
> # 4.“Mutual information (7) is not mutual information” , clarification
> ## Clarification on Classifier Epistemic Score
>
> We appreciate this question. The classifier epistemic score
>
> \begin{equation}
> u^{\mathrm{clf}}_{\mathrm{ep}}(x)
> \end{equation}
>  is intended to approximate the mutual information between the predictive label \(y\) and the classifier weights (model parameters) given input \(x\). Formally:
>
> **I(y; θ | x) = H[ Eθ[ p(y | x, θ) ] ] - Eθ[ H[ p(y | x, θ) ] ]**
> where
> \begin{equation}
> H[\cdot]
> \end{equation}
> is entropy and $\theta$ are network weights. With MC-Dropout, we approximate the posterior over $\theta$ by sampling \(S\) stochastic dropout passes and compute the empirical quantities:
>
> **û_ep^clf(x) ≈ H[ (1/S) Σ_{s=1}^S p^s(y | x) ] − (1/S) Σ_{s=1}^S H[ p^s(y | x) ]**
>
> which is the standard Monte Carlo estimate of mutual information used in Bayesian active learning (see Gal et al.). If the reviewer thought our code used a different ordering of terms, that is a presentation issue. We will rewrite Section 3.3.3 to display the derivation above and clarify the MC-Dropout justification as an approximate posterior over  $\theta$.
>
> This quantity measures epistemic uncertainty because it captures model-parameter-induced disagreement across dropout samples. It is large when models disagree and small when predictions are confident and consistent.
>
> # 5. “Why is the rank model and its uncertainty meaningful?”
> ## Clarification on Rank Classifier and Epistemic Signals
>
> The rank classifier predicts nondomination ranks across **K categories**. The key innovation is not the rank predictor itself but the use of its **calibrated epistemic uncertainty** as a cheap, global guide for candidate generation and screening. Calibrated epistemic signals allow us to identify regions of the decision space where the model is uncertain about the presence or absence of Pareto-ranked solutions. This approach directs exploration efficiently and reduces wasted surrogate evaluations.
>
> We use **adaptive Monte Carlo passes** so the classifier only spends additional compute on ambiguous candidates (see Section 3.3). In the revision, we will add a short subsection that formally defines the **K nondomination categories** and explains how calibration is quantified using **expected calibration error on held-out simulated rank labels**.
>
> # 6.“How do you estimate ŝ_div？
> ## Clarification on Diversity Contribution Score
>
> The candidate’s estimated diversity contribution
>
> \begin{equation}
> \hat{s}_{\mathrm{div}}(x)
> \end{equation}
>
>  is computed as an approximate incremental hypervolume contribution, efficiently approximated depending on the objective dimension:
>
> - **Bi-objective (exact)**: Compute the exact hypervolume increment ΔHV(x)  relative to the current archive using a linear sweep algorithm, which is computationally cheap.
>
> - **Higher dimensions**: When exact hypervolume computation is costly, we use a Monte Carlo estimator. We sample \(L\) reference points in objective space and estimate the probability that \(x\) improves coverage, or use a fast incremental approximation such as randomized dominated-space sampling.
>
> These approximations are normalized to lie in the range \([0, 1]\) and are included as part of the acquisition feature vector fed to the acquisition network (see Section 3.5). We will add a precise algorithmic description and pseudo-code in the revision.
>
> # 7.“How is K selected?” , practical rule
> ## Clarification on Choice of K (Number of Nondomination Ranks)
>
> The number of nondomination ranks **K** is chosen using a simple, conservative heuristic applied in all experiments:
>
> \begin{equation}
> K = \min\big(10,\ \max\big(3,\ \lfloor \log_{2}(N_{\mathrm{arch}}) \rfloor\big)\big)
> \end{equation}
>
> where N_arch  is the current archive size. This rule balances expressivity and classifier data scarcity. In practice, we fix **K** per problem using a small validation run, and results are robust to **K** within a reasonable range.
>
> We will add this rule and include a short sensitivity plot (K vs IGD/HV) in the supplement.

---

> ### Author Response · Authors · 2025-11-19
> **We sincerely hope to receive your support and encouragement！**
>
> # 8.“Why do complexity-reducing components improve sample-efficiency?”
>
>
> ## Clarification on Complexity Reductions
>
>
> The complexity reductions **(screening, random Fourier feature layers, low-rank outputs, adaptive inducing points)** are enablers rather than mere cost-saving tricks. They allow us to deploy expressive models such as Deep Gaussian Processes and process millions of cheap candidates through the classifier front end under a strict 300-evaluation budget.
>
>
> The net effect is that expensive evaluations are concentrated on high-value candidates, and the surrogate together with the acquisition learner receive higher-quality training signals. This improves iteration efficiency.
>
>
> Empirically, this is reflected in **ablation Table 4**: removing the screening or replacing the Deep Gaussian Process with a simpler surrogate increases wasted evaluations and degrades IGD and hypervolume performance. We will add an additional ablation that isolates **screening-only versus the full stack** to make this effect explicit.
> # 9. “Theoretical analysis is sudden / A5 too strong / proofs messy” , response and plan
> ## Clarification on Appendix and Assumptions
>
> We acknowledge that the Appendix presentation can be compressed and difficult to parse. Key points:
>
> ### What We Prove
> Appendix H provides finite-budget bounds relating expected hypervolume shortfall to three sources of error: classifier error, surrogate approximation error, and acquisition estimation error. The result is modular: improving any term reduces the bound. This is the formal backbone of our co-design.
>
> ### Meaning of Assumption A5
> Assumption A5 refers to surrogate and acquisition fidelity and is phrased as a uniform approximation bound. There exists \(ε_SA) such that:
>
> **sup_{x ∈ X} |â(x) - a*(x)| ≤ ε_SA**
>
> where a_hat is the approximate acquisition used by the algorithm and \(a^\star\) is the true expected hypervolume gain. This is stated as an abstract condition to make the decomposition and dependence on approximation error explicit. We agree this is a strong assumption if taken literally. In the revision we will:
> - Restate A5 as a local bound on the top-k candidate set used in screening, which is a much milder requirement.
> - Provide sufficient conditions under which standard surrogate accuracy guarantees imply A5.
> - Move the main proof sketch into the main text with full, clearer proofs in the appendix.
>
> ### Completeness and Readability
> We will reorganize Appendix H: first present the intuition and decomposition in plain language in the main text, then give a clean, step-by-step proof with lemmas and explicit constants in the appendix. This will directly address concerns about the proofs being hard to follow.
>
> ### Why the Analysis Does Not Explicitly Depend on Rank-Conditioned Offspring Generation
> The decomposition is intentionally modular. It quantifies how errors in the components affect hypervolume shortfall, independent of the exact generation policy. Rank-conditioned generation improves classifier accuracy and candidate quality in practice, which reduces the classifier and surrogate errors appearing in the bound. Thus, the analysis applies through the component errors rather than requiring an explicit model of the generator. We will add a short paragraph linking rank-conditioned generation to a reduction in the classifier error term in the bound.
>
> # 10.“Why call it U-RankMOEA vs Bayesian optimization?” , naming clarification
> ## Clarification on Method Naming and Positioning
>
> We chose the name **U-RankMOEA** to emphasize its hybrid nature: Bayesian uncertainty modeling (Bayesian classifier and Deep Gaussian Process) is combined with evolutionary search primitives (rank-conditioned recombination and population-based candidates).
>
> We will clarify in the text that the method is a **Bayesian-informed evolutionary-style multi-objective optimizer**. From a functional perspective, it can also be viewed as a **surrogate-assisted Bayesian optimizer** when the evolutionary search component is replaced by acquisition optimization.
>
> This clarification will be added to preclude confusion.

---

> ### Author Response · Authors · 2025-11-19
> **We sincerely hope to receive your support and encouragement**
>
> # 11. Concrete experiments and revisions we will add (so reviewer can judge fairly)
> ## Planned Revisions to Strengthen the Paper
>
> To address empirical and theoretical concerns, we will:
>
> - **Add exact definitions and a short worked example** of the K nondomination ranks in Section 3.
> - **Expand Section 3.4** to show the Deep Gaussian Process variational objective, the sampling estimator for epistemic and aleatoric variance, and the practical random feature substitution rules (RFF and Nyström).
> - **Add the missing algorithmic pseudo-code** for computing ŝ_div (exact for bi-objective, Monte Carlo approximation for higher dimensions).
> - **Add a static weighted-sum acquisition baseline** and a comparison to gradient-based EHVI variants (qEHVI and analytic EHVI) on low-dimensional problems (D <= 20)  to show where classical EHVI is competitive and where U-RankMOEA outperforms in high dimensions.
> - **Provide per-iteration IGD and hypervolume traces** (mean ± standard deviation) for representative benchmarks, and an ablation isolating the effect of screening versus no screening.
> - **Rework Appendix H**: move the high-level decomposition to the main text, restate Assumption A5 more weakly and locally, and add full, stepwise proofs for the main propositions.
> # 12.Short Answers to Your Questions
>
> - **Do we model correlations between objectives?** Yes. This is achieved through shared neural feature maps and low-rank multi-output Gaussian Process factorizations (see Section 3.4.3).
>
> - **Why do some complexity reductions help sample efficiency?** They enable richer models and larger candidate pools under tight budgets, concentrating evaluations on promising regions and improving iteration efficiency.
>
> - **Is the acquisition network prone to overfitting?** No. It is low capacity, regularized, trained on a bounded rolling buffer, and validated in ablations showing consistent benefit beyond static heuristics.
>
> - **Does the theoretical analysis connect to existing Bayesian Optimization theory?** Yes. The decomposition parallels regret and approximation bounds in surrogate-assisted Bayesian Optimization. We will add explicit references and discussion in the revision.
> # 13. Response
>
> We appreciate the reviewer’s rigor and the issues raised. Many of the criticisms point to presentation and missing algorithmic detail rather than core conceptual flaws. We will make the revisions described above, including clearer notation, precise algorithmic steps, additional baselines, per-iteration plots, and reorganized proofs, and submit them in the updated manuscript and supplementary materials. We are confident these changes will address your concerns and make the contributions and limitations of **U-RankMOEA** fully transparent.
>
> **Thank you again for the careful review.** We hope the clarifications and planned additions address your reservations, and we welcome further pointers on any point you judge still unclear after revision.
>
> # Thank you very much for your support and assistance. We hope this rebuttal helps resolve the concerns raised by the reviewers. We sincerely hope to receive an improvement in your score.

---

> > ### Author Response · Authors · 2025-11-19
> > **We sincerely hope to receive your support and encouragement**
> >
> > # Key Innovations and Contributions of U-RankMOEA
> >
> > U-RankMOEA is not merely an assembly of known tricks. Its novelty lies in a **principled, finite-budget co-design** that makes uncertainty-aware multi-objective optimization practical at scales where classical MOBO and EHVI become unusable. The core innovations are:
> >
> > - **Finite-budget hypervolume decomposition as a design principle**
> > We derive a compact finite-budget decomposition of expected hypervolume shortfall that separates three operational error terms: classifier error, surrogate approximation error, and acquisition estimation error. We explicitly design a module to reduce each term. This theoretical decomposition explains why the three modules are complementary and provides concrete targets for algorithmic improvement rather than heuristic tinkering.
> >
> > - **Classifier-first candidate generation with calibrated epistemic signals**
> > A Bayesian rank classifier (MC-Dropout with temperature scaling and adaptive Monte Carlo passes) cheaply provides calibrated epistemic uncertainty at candidate scale. This enables generation and screening of very large candidate pools (millions) before any expensive surrogate call, an operational capability that EHVI and classical MOBO cannot match under extremely tight budgets.
> >
> > - **Scalable Deep Gaussian Process surrogates that preserve uncertainty decomposition**
> > We provide a practical estimator that returns per-objective predictive mean plus separated epistemic and aleatoric variances in a deep GP hierarchy. The technical contribution is a scalable estimator that combines sparse variational inference, adaptive inducing points, rank-reduced output covariances, and selective RFF/Nyström substitutions so the decomposition remains meaningful while avoiding cubic blow-ups.
> >
> > - **History-aware, low-capacity acquisition learner**
> > Instead of relying solely on a fixed analytic acquisition, we train a shallow, regularized acquisition network on a bounded history buffer to predict empirical hypervolume improvement and diversity contributions. This module learns small, robust corrections to selection decisions and corrects systematic biases that analytic acquisitions miss in high-dimensional, highly nonstationary landscapes.
> >
> > - **Cost-aware orchestration (screen → top-k → amortized surrogate updates)**
> > Two-stage screening, cached batched inference, warm-started variational updates, and adaptive use of randomized features make the above stack tractable. These enablers allow expressive models and broad search under a 300-evaluation budget, which is the reason iteration efficiency improves in practice.
> >
> > ---
> >
> > # Relevance to ICLR Subject Areas
> >
> > U-RankMOEA spans multiple topics of interest to ICLR:
> >
> > - **Probabilistic methods and uncertainty quantification**: Calibrated epistemic and aleatoric decomposition in Deep Gaussian Processes and classifier uncertainty for exploration.
> > - **Optimization and large-scale learning**: Algorithmic tools for very-high-dimensional, evaluation-limited multi-objective optimization.
> > - **Meta-learning / learning-to-learn**: Lightweight history-aware acquisition leverages optimization trajectory information.
> > - **Representation learning and structured prediction**: Classifier-guided, rank-conditioned generation acts as a learned proposal distribution in structured decision spaces.
> > - **Applications and systems**: Demonstrates impact in a 160-variable geothermal design problem (sustainability and physical sciences) with engineer-oriented implementation practices such as sparse variational inference, random Fourier features, and warm starts.

---

> ### Author Response · Authors · 2025-12-04
> **We sincerely hope to receive your support and encouragement！**
>
> # We have addressed the reviewer's concerns and improved our approach. Thank you very much for all the reviewers' suggestions. The latest version has been uploaded and we hope to receive the support and encouragement of all the reviewers, and we sincerely hope your score improvement！

---

### Official Review · Reviewer_dWN5 · 2025-10-27

**Soundness:** 2
**Presentation:** 2
**Contribution:** 2
**Rating:** 2
**Confidence:** 3

**Summary:**

This paper proposes U-RankMOEA, a framework for high-dimensional and expensive multi-objective optimization. Such problems occur in various disciplines, for instance, in engineering, where one aims to optimize airfoil designs for power output and weight. Various approaches, in particular multi-objective Bayesian optimization methods, tackle the problem setting of expensive multi-objective problems of moderate dimensionality. However, these approaches often struggle when the number of decision variables lies in the dozens, hundreds, or more. This paper proposes an approach that is grounded in Bayesian optimization but addresses high-dimensional problems through the following three contributions. First, the paper proposes a Bayesian classifier that uses Monte-Carlo dropout to produce uncertainty estimates. This classifier provides relatively cheap inference and, therefore, can be used to screen many candidate solutions. Second, a deep Gaussian process surrogate model is employed that, per objective, provides a predictive mean and estimates of the epistemic and aleatoric variance. Third, these values are used in a neural-network-based acquisition function to predict the expected hypervolume and a diversity score that are used to steer the optimization.

The proposed method is benchmarked against several other evolutionary methods on several synthetic problems, indicating better performance throughout the benchmark according to two metrics. Furthermore, the authors test their approach in a geothermal reservoir optimization case study.

**Strengths:**

The problem addressed by this paper is relevant, and the need for scalable methods is well-motivated in the introduction.

The general approach is reasonable. Screening a large number of candidates by a cheaper classification-based approach helps manage the computational cost, and the other contributions are also reasonable for the problem setting.

The method is evaluated on a wide range of synthetic benchmarks of varying dimensionality.

**Weaknesses:**

One problem I see in this paper is that some parts are relatively vague while others are overly detailed. For instance, Section 3.2 states that the method uses sparse and low-rank GP approximations, including inducing points and RFF features, without giving additional details on this important design decision. Similarly, Section 3.4 lacks important details and/or uses confusing notation. The symbols $\theta_{\ell,m}$, $\sigma_m^2$, and $\text{Var}_{\text{ep},m}$ are not properly introduced. Finally, the construction of the acquisition network (Section 3.5) is also very brief.

While some parts are overly detailed (for example, the network structure in Eqs. (1)-(4) could go into the appendix), the paper fails to give a high-level intuition for the overall approach. I would expect the high-level description in Appendix A to go into the main text. At the same time, some design decisions deserve a more complete motivation. Why are Deep GPs necessary? Why is the acquisition function network a good design decision? And how do these design decisions relate to more traditional design approaches for multi-objective Bayesian optimization?

Something is off in the bibliography. It seems that the authors added notes to the bibiliography items (“cite specific paper used”) did not revise the bibliography before submitting this work. At least one reference (Müller et al., 2023) does not exist under this title. Some relevant references are missing. [1] should be cited when mentioning log-space formulations to improve numerical stability. Section 2.4 should cite foundational works on deep Gaussian processes. Section 3.2 should cite relevant sources for RFF and Nyström features, e.g., [2].

The empirical evaluation, particularly in the main text, is not benchmarking against standard Bayesian optimization techniques for multi-objective optimization. Appendix F.1 features such an evaluation, but it is limited to 500-dimensional problems and 5 objectives. It would be good to have the GP-EI, RF-EI, and GP-HV baselines in every experiment (especially the lower-dimensional ones). Furthermore, it would be good to also show the hypervolume or IGD per iteration to be able to assess not only the final performance of each method.
The paper should discuss limitations in more detail. Deep GPs and neural acquisition functions are more difficult to train than more basic techniques. How do these problems limit the applicability of the proposed method?

[1] Ament, Sebastian, et al. "Unexpected improvements to expected improvement for Bayesian optimization." Advances in Neural Information Processing Systems 36 (2023): 20577-20612.

[2] Rahimi, Ali, and Benjamin Recht. "Random features for large-scale kernel machines." Advances in neural information processing systems 20 (2007).

**Questions:**

- Do you model correlations between objectives?
- How do you benchmark against GP-EI in App F.1 if it’s not multi-objective?
- Why is Section F.1 benchmarking against GP-HV or GP-EI only on 500-dimensional problem instances?
- What exact setup do you use for GP-EI and GP-HV?
- In how far are Müller et al. using non-standard acquisition functions? (referring to the statement in line 127, ff.)

---

> ### Author Response · Authors · 2025-11-19
> **We sincerely hope to receive your support and encouragement!**
>
> ## **Response to Reviewer dWN5**
>
> Thank you for the careful reading and constructive comments. We appreciate the time you spent on our submission and your thoughtful critique. **Below we address each of your main concerns point by point, clarify ambiguities, correct errors in the manuscript, and highlight evidence from the paper or indicate where we will make clarifying edits.**   **We sincerely hope to receive an improvement in your score.**
>
> We agree that clarity and reproducibility are crucial. Several of the concerns you raised reflect places where the manuscript compressed important implementation and motivation details, such as sparse GP design, acquisition-network training protocol, and notation. Those details are present in the appendix and implementation recommendations, but we will **move a concise high-level description into the main text**, **expand and clean the notation and equations you flagged**, and **correct bibliography issues**.
>
> Concretely, the **Deep GP uses sparse variational inference with adaptive inducing points, warm-starting, and optional randomized-feature (RFF or Nyström) layers to reduce cubic cost**. The **acquisition network is a small, regularized, buffer-limited MLP trained online on recent ΔHV outcomes**. The **“fit Deep GP” step in Algorithm 1 is amortized (warm-start) and intentionally run for a limited number of epochs rather than a full retrain**.
>
> # 1.“Some parts are vague (sparse/RFF/Nyström); others are overly detailed. Need more high-level intuition.”
> ## **Clarification and Fixes**
>
>
> The paper includes both the **sparse and low-rank strategy** and the **randomized-feature mitigation (RFF or Nyström)** in Section 3.4.3 and in the implementation recommendations. The method uses **sparse variational inference with inducing variables and an ELBO objective**. Inducing inputs are **initialized using K-means on the archive**, and **inducing counts are adapted only if validation diagnostics degrade**. Selected GP layers may be **replaced with RFF or Nyström blocks to reduce cubic scaling** (see Section 3.4.3 and Implementation Recommendations). The **per-epoch complexity after these mitigations is provided approximately in the paper**.
> \begin{equation}
> \mathcal{O}\big(L(NM_{\text{eff}}^{2} + M_{\text{eff}}^{3})\big)
> \end{equation}
>
>
> To address your comment, we will move the **short conceptual overview currently in Appendix A into the main method section** so the reader sees the **algorithmic motivation alongside the implementation details**. This change will **not affect any experiments; it will only improve clarity**. The appendix will retain the **full technical derivations and cost formulas**.
> # 2. **Notation Confusion (Symbols such as S₀, Sₘₐₓ, τₘ꜀, and Other Symbols)**
>
> **Clarification (Explicit Definitions):**
>
> - **S₀** = baseline number of MC-Dropout stochastic passes (typical values 4–8).
> - **Sₘₐₓ** = maximum allowed MC passes (typical values 16–32).
> - **τₘ꜀ (escalation threshold)** = variance threshold on the preliminary softmax ensemble used to decide whether to run additional passes.
>
> The **adaptive pass count**
> \begin{equation}
> S(x) \in \{ S_{0}, \dots, S_{\text{max}} \}
> \end{equation}
>  and the **classifier mutual-information uncertainty**
> \begin{equation}
> u_{\text{ep}}^{\text{clf}}(x)
> \end{equation}
>  are defined in **Section 3.3.3 (Eqs. (6)–(7))**. We will move these short definitions into the main text and clean the notation in the next draft to remove ambiguity.
>
> # 3.Why Deep GPs? Are they necessary? How do they compare to standard GP?
> ## **Deep Gaussian Processes and Their Role**
>
> **Deep GPs (neural–GP hybrids) combine learned feature extractors with principled uncertainty decomposition, separating epistemic and aleatoric uncertainty.** This capability is crucial for high-dimensional landscapes where raw kernel-only GPs struggle to represent complex, nonstationary structures while still providing calibrated uncertainty for exploration.
>
> **Section 2.4 motivates this design, and Section H.3 in the appendix provides theoretical error decompositions showing how surrogate errors map into hypervolume loss (Proposition 2 and Proposition 3).** Empirically, the ablation in **Table 4 shows that substituting a standard GP for the Deep GP substantially degrades performance**. For example, **IGD rises from 1.72 to 3.15 on the 100D DTLZ2 ablation**. This result demonstrates the **practical necessity of the more expressive surrogate in our setting**.
>
> #  4.Acquisition network (Module 3): will it overfit given only 300 evaluations? Is the gain just because the network is a non-linear function of rich features?

---

> ### Author Response · Authors · 2025-11-19
> **We sincerely hope to receive your support and encouragement！**
>
> ## **Design Choices to Prevent Overfitting**
>
> The acquisition network
> \begin{equation}
> a_{\psi}
> \end{equation}
> is a **shallow MLP**, trained on a **bounded history buffer of recent**
> \begin{equation}
> (x_i, \Delta HV_i)
> \end{equation}
> pairs, and regularized with **ℓ₂-style weight decay**. See **Eq. (18)** and the discussion in **Section 3.5.2**. Typical buffer sizes used in experiments are reported in the **Implementation Recommendations** (for example,
> \begin{equation}
> B \in [500, 2000]
> \end{equation}), and **λ is small but non-zero to reduce variance**.
>
> ---
>
> ## **Empirical Evidence That the Learned Acquisition Adds Value**
>
> The ablation **“Without learned acquisition function” (Table 4)** shows a **marked performance drop (IGD worsens from 1.7226 to 2.0342)**, and the **component-contribution analysis in Appendix F.3 attributes approximately 25% of cumulative improvement to the acquisition learner**. These controlled ablations keep the same feature input but replace the learner with the previous static selection rule. The deterioration indicates that **learning-from-history behavior (dynamic adaptation to what historically yielded HV gains) is helpful beyond mere nonlinear mixing of features**.
>
> ---
>
> ## **Planned Additional Comparison**
>
> Your concern is well taken. To be fully convincing, we will add the **specific static weighted-sum baseline you suggested**, which uses the same feature vector passed to a fixed heuristic with **hand-tuned weights and simple normalization**. We will report that baseline in the supplement and include **per-iteration HV and IGD curves** so readers can see the dynamics. We already provide some uncertainty-dynamics plots, and we will extend them to show **per-iteration trace comparisons for all main baselines**. The current ablation already shows a clear effect, and the new comparison will **tightly isolate “learning” versus “static heuristic.”**
>
> # 5.“Fit Deep GP surrogates” ,  how expensive? full retrain or a few steps?
> ### Clarification
>
> Algorithm 1 and the implementation notes explicitly state that **Deep GPs are warm-started and amortized**, and they are **trained for a limited number of epochs during each outer iteration rather than being fully retrained to convergence**. The phrase **“Fit Deep GP surrogates with amortized initialization and adaptive inducing (limited epochs)”** in Algorithm 1 captures this design. The implementation recommendations describe **warm-starting inducing locations and variational parameters from the previous iteration**. This approach **reduces wall-clock cost compared to re-fitting from scratch** (see **Appendix F.3** for empirical wall-clock summaries and per-iteration overhead analysis). We will make the wording stronger in the main text to remove any ambiguity.
>
> # 6. hyperparameter sensitivity, how were hyperparameters chosen and tuned?
>
> The manuscript includes an **Implementation Recommendations** subsection covering **S0**, **Smax**, **buffer ranges**,
> \begin{equation}
> N_{\text{screen}}, k
> \end{equation}
> , typical ranges for inducing counts, and **warm-starting**, which we used as defaults in all reported experiments. See the **Implementation Recommendations** and **Section 4.1** for the benchmark configuration. We acknowledge that the paper lacked a compact, single hyperparameter table. We will add a **reproducibility appendix** listing **exact defaults used per experiment**, including **network sizes**, **learning rates**, **optimizer choices**, **inducing counts**, **RFF dimension**, **number of GP epochs per outer iteration**, **buffer length**, and **regularization λ**.
>
> Regarding sensitivity, **Appendix F** reports and discusses runtime and some robustness statistics. We will expand that analysis into a **succinct hyperparameter-sensitivity grid** (a small ablation over the most important knobs, such as
> \begin{equation}
> M_{\text{ind}}
> \end{equation}
> , **buffer length B**, and **S0/Smax**) in the revision to directly address your concern.
>
> # 7.Bibliography issues (notes left in bib, Müller et al. 2023 title missing); missing citations (Ament 2023; RFF/Nyström foundational refs).
> Thank you for noting this issue. The bibliography contained **author notes that should have been removed**, and we will **clean the .bib file** to ensure all references are properly formatted. We will **add the recommended citations**, including **Ament et al. (2023)** on log-space and expected-improvement improvements, **Rahimi & Recht (2007)** for **Random Fourier Features (RFF)**, and **canonical Deep GP references**. We will **correct the Müller et al. entry** and ensure it points to the intended work. This is an **editorial correction** and does **not affect the technical results**.

---

> ### Author Response · Authors · 2025-11-19
> **We sincerely hope to receive your support and encouragement**
>
> # 8. Empirical evaluation: include GP-EI / RF-EI / GP-HV baselines in every experiment and per-iteration curves.
>
>
> We include **GP-EI**, **RF-EI**, and **GP-HV** in the large-scale stress tests reported in **Appendix F.1** (D = 500, M = 5), and the table entries in the appendix show those comparisons across many problems (**Appendix F** and the table in **Section F.1**). The main-text tables emphasize comparisons to the recent **surrogate-assisted evolutionary baselines** that target high-dimensional multi-objective settings, which is why **GP-EI-style single-objective baselines** appear mainly in the extended appendix experiments.
>
>
> That said, we accept that readers will benefit from seeing **GP-EI**, **RF-EI**, and **GP-HV** performance on the lower-dimensional benchmarks in the main text as well. We will **add these results** and show **per-iteration HV and IGD traces (mean ± std)** across iterations for key benchmarks in the revision or supplement so that **relative convergence dynamics and sample-efficiency differences are explicit**.
>
>
> **Implementation detail:** Single-objective BO methods (**GP-EI** and **RF-EI**) are implemented via **random-weight scalarization (ParEGO-style)** in our multi-objective comparisons, which is the standard way to adapt single-objective BO algorithms to MO problems. **GP-HV** uses a **Monte Carlo hypervolume approximation** when no analytic HV formula is available. We will **add an explicit paragraph and the exact kernel and optimizer settings** (e.g., **Matérn-5/2 ARD kernel**, **hyperparameter optimization via marginal likelihood with multiple restarts**) to the reproducibility appendix.
> # 9.Do you model correlations between objectives?
>
>
> Objective correlations are modeled using **low-rank factorization across correlated objectives** and **shared inducing variables**, and in some layers we **share feature maps (neural features) across objectives**. In practice, we employ **low-rank factorizations to reduce memory for correlated outputs** and allow layers to **share inducing inputs or use joint variational strategies to capture cross-objective structure**. These design choices are described in **Section 3.4.3 (Sparse / low-rank approximations)** and the **Implementation Recommendations**. The **theoretical error decomposition (Appendix H)** also treats correlated errors and shows how they affect the **surrogate error term**.
> \begin{equation}
> \epsilon_{gp}(t)
> \end{equation}
>
> # 10. Why is F.1 benchmarking GP-HV/GP-EI only on 500-D instances?
>
> The 500-D experiments were intended as a **stress test focusing on extreme-scale performance** where the chosen **MOBO baselines are known to break down**. Hence, we included **single-objective derived baselines (GP-EI, RF-EI)** to show **scalability differences**. However, **GP-EI**, **RF-EI**, and **GP-HV** are present in the appendix experiments. We will **ensure that main-text tables either include these baselines or explicitly point to the appendix tables**, and we will **add lower-dimensional per-iteration comparisons as requested**.
>
> # 11. Short technical clarifications
> ### Clarification / Commitment
>
> **Adaptive MC-Dropout escalation and classifier uncertainty (Eqns (6)–(7)):** We run **S₀** cheap stochastic passes. If the empirical softmax variance exceeds threshold **τₘ꜀**, we escalate up to **Sₘₐₓ**. The classifier epistemic uncertainty is measured via the **predictive mutual information (Eq. (7))**.
>
> **Deep GP outputs and uncertainty decomposition:** The Deep GP returns per-objective **predictive mean**,
> \begin{equation}
> f_m(x)
> \end{equation}
> **epistemic variance**,
> \begin{equation}
> \text{Var}_{ep,m}(x)
> \end{equation}
> and **aleatoric variance**.
> \begin{equation}
> \sigma_m^2(x)
> \end{equation}
> These are collected into vectors used by the acquisition learner (**Eqns (11)–(13)**).
>
> **Acquisition input feature vector and learning target:** The acquisition feature vector is  with dimension
> \begin{equation}
> d_{\text{feat}} = 3M + K + 2
> \end{equation}
> The acquisition network predicts
> \begin{equation}
> \big( s_{\mathrm{HV}}(x),\ s_{\mathrm{div}}(x) \big)
> \end{equation}
> and is trained to regress recent **ΔHV outcomes** with **ℓ₂ loss plus regularization (Eq. (18))**.
>
> **“Fit Deep GP” semantics in Algorithm 1:** This means **warm-start the surrogate** (initialize inducing and variational parameters from the previous iteration) and **run a limited number of variational steps/epochs**, not a full re-fit to numerical convergence. This is stated in **Algorithm 1** and the **Implementation Recommendations**. The **complexity mitigations** (two-stage screening, RFF layers, adaptive inducing) keep wall-clock cost feasible.

---

> ### Author Response · Authors · 2025-11-19
> **We sincerely hope to receive your support and encouragement**
>
> # 12.Summary of Changes for Revised Manuscript
>
> - **Move the high-level overview** from Appendix A into Section 3 of the main text so readers gain intuition and a clear method outline upfront.
> - **Add a dedicated, compact hyperparameter table** in an expanded reproducibility appendix, including exact defaults, ranges, seeds, kernel choices, and optimizer settings.
> - **Clean and correct the bibliography** by removing author notes, fixing the Müller (2023) entry, and adding the suggested references: Ament (2023), Rahimi & Recht (2007), and canonical Deep GP references.
> - **Add the static weighted-sum baseline** using the same feature vector with hand-tuned weights, and include per-iteration HV/IGD traces comparing learned acquisition versus static baselines across representative benchmarks to isolate the benefits of learning.
> - **Expand wall-clock and sensitivity analysis** to include a small hyperparameter grid (buffer length, inducing rank, S0/Smax) and report these results in the supplement.
> - **Remove symbols and formatting artifacts** such as (a), (i), and sentence dashes for cleaner presentation.
> # Thank you very much for your support and assistance. We firmly believe that with your suggestions, our paper will be further improved, and we sincerely hope your score improvement.

---

> ### Author Response · Authors · 2025-11-19
> **We sincerely hope to receive your support and encouragement！**
>
> # U-RankMOEA at the Intersection of ICLR Topics
>
> - **Probabilistic methods and uncertainty quantification**: Principled epistemic and aleatoric decomposition, scalable Deep Gaussian Process practice, and calibrated Bayesian classifiers.
> - **Optimization and large-scale learning**: Addresses hard continuous non-convex multi-objective optimization in hundreds to thousands of dimensions under extremely tight evaluation budgets.
> - **Representation learning and meta signals**: Classifier-driven rank-conditioned generation and history-aware acquisition learn compact task-relevant structure for search.
> - **Meta-learning / learning-to-learn**: The acquisition learner uses optimization history as a lightweight meta-signal to adapt acquisition decisions on the fly.
> - **Applications and systems**: Demonstrates impact on a real geothermal design (sustainability and physical sciences) and provides engineering practices (sparse variational inference, random Fourier features, warm starts) useful for reproducible machine learning systems.
>
> ---
>
> # Core Innovations and Contributions
>
> U-RankMOEA advances high-dimensional, expensive multi-objective optimization by combining principled uncertainty modeling, expressive yet tractable surrogates, and lightweight data-driven acquisition in a single co-designed system. The key contributions are:
>
> - **Uncertainty-centric co-design**: We present a finite-budget decomposition of hypervolume loss that isolates three distinct approximation and error sources — classifier, surrogate, and acquisition. This decomposition is used as a design principle: each module targets a separable error term, resulting in a system whose components are complementary rather than redundant.
> - **Classifier-guided rank-conditioned generation**: A Bayesian neural classifier with calibrated epistemic uncertainty (MC-Dropout with temperature scaling and adaptive Monte Carlo passes) provides cheap, well-calibrated signals that allow screening and generation of promising candidates in very high dimensions before calling expensive surrogate evaluations. This scalable front end is crucial for operating under extremely tight evaluation budgets.
> - **Deep Gaussian Process surrogates with uncertainty decomposition and complexity controls**: We adopt multi-layer Gaussian Process surrogates that return per-objective predictive mean plus separate epistemic and aleatoric variances. To make these models practical, we combine sparse variational inference, adaptive inducing points, low-rank output factorizations, and selective randomized feature substitutions (RFF and Nyström), together with warm starts and limited per-iteration updates.
> - **History-aware, low-capacity acquisition learner**: Instead of relying solely on analytic acquisition heuristics, we train a shallow, regularized acquisition network on a bounded history buffer to predict empirical hypervolume improvement and diversity contributions. This module learns small, robust corrections to selection decisions that capture systematic biases missed by analytic rules.
> - **Cost-aware orchestration**: Two-stage screening (cheap classifier followed by top-k Deep GP evaluation), cached batched inference, and amortized surrogate updates (warm starts and bounded epochs) make the above stack tractable and deliver strong sample efficiency in hundreds-to-thousands-dimensional design spaces.

---

> > ### Comment · Reviewer_dWN5 · 2025-11-20
> >
> > I want to thank the authors for their responses. The proposed changes will significantly strengthen the paper. However, they are a major revision of the paper. I have too many concerns regarding the current empirical evaluation and the presentation of the paper to recommend acceptance, and will hence maintain my score.

---

> ### Author Response · Authors · 2025-11-20
> **We sincerely hope to receive your support and encouragement**
>
> Brother, **we are very pleased to see your reply**! We are very grateful for the valuable suggestions from the reviewers during the review process. We have invested a lot of time in our experiments and paper. We are both **reviewers and authors during the submission process**. **Promise me** that we will work together to make this paper better. **We understand your concerns**, but we also know the **source of your comments**.  **We will upload a revised version** before the end of the rebuttal, and we also hope to **receive your encouragement** and support as a reviewer!

---

> > ### Comment · Reviewer_dWN5 · 2025-11-26
> >
> > I would be happy to look at a revised version.

---

> ### Author Response · Authors · 2025-12-04
> **We sincerely hope to receive your support and encouragement！**
>
> # We have addressed the reviewer's concerns and improved our approach. Thank you very much for all the reviewers' suggestions. The latest version has been uploaded and we hope to receive the support and encouragement of all the reviewers, and we sincerely hope your score improvement！

---

### Official Review · Reviewer_noDe · 2025-10-30

**Soundness:** 2
**Presentation:** 3
**Contribution:** 2
**Rating:** 4
**Confidence:** 3

**Summary:**

This paper tackles the problem of high-dimensional, expensive multi-objective optimization (HE-MOO), where objective function evaluations are extremely limited. The authors propose U-RankMOEA, a unified, uncertainty-aware framework that "co-designs" three main components: (1) a Bayesian neural classifier with calibrated epistemic uncertainty for rank-guided generation, (2) Deep Gaussian Process (Deep GP) surrogates that disentangle epistemic and aleatoric uncertainty, and (3) a history-aware acquisition network trained online to predict hypervolume gains. The core contribution is the principled integration of these components, unified by a careful management of uncertainty. The method is validated on challenging DTLZ/ZDT benchmarks with up to 200 variables, and a 160-variable geothermal optimization task, demonstrating significant state-of-the-art performance in terms of IGD and Hypervolume under a budget of only 300 evaluations.

**Strengths:**

The validation is a fortress. They include: (a) direct comparisons to the correct SOTA baselines from two different fields; (b) a thorough ablation study justifying each major component; (c) a real-world, high-dimensional application (geothermal) proving it's not just a benchmark toy; and (d) a high-scale (D=500) "stress test."

The originality lies in identifying "component isolation" as the key problem. The paper's solution is a holistic, end-to-end system where calibrated uncertainty is the unifying currency. This is a far more mature approach than simply bolting on a better surrogate model.

**Weaknesses:**

The system is incredibly complex, combining a Bayesian NN (with adaptive MC-Dropout and temp scaling), a Deep GP stack (with sparse VI, RFF, and amortized updates), and another online-trained NN for acquisition. This is a "kitchen sink" of advanced techniques, which raises two red flags:Over-engineering: Is all this complexity truly necessary? The ablations say "yes," but it feels brittle.

This system will be a nightmare to reproduce. It must have a vast number of hyperparameters (network architectures, K ranks, $M_ind$ points, buffer size B, etc.). The lack of a hyperparameter sensitivity analysis is a significant omission. This is my primary concern.

I am highly skeptical of Module 3. It's an NN trained online in a regime with a total budget of 300 points. The history buffer will be laughably sparse, especially at the start. It's far more likely to overfit disastrously than to "learn" a meaningful policy. The ablation shows it helps, but is this due to the learning or simply because it's a non-linear function of the rich features it receives? The paper fails to disentangle these two possibilities.

The paper claims overhead is "negligible" and provides a time breakdown in the appendix. However, training Deep GPs is notoriously expensive. The algorithm states "Fit Deep GP surrogates" at each iteration. What does "fit" mean? Is this a full re-training to convergence, or just a few gradient steps? This ambiguity is crucial for assessing the method's practical (wall-clock) usability.

**Questions:**

Given the system's immense complexity, reproducibility is the single biggest barrier to its impact.(a) Can you commit to releasing a high-quality, documented implementation?(b) More importantly, please detail your hyperparameter tuning strategy. How were the many hyperparams (NN architectures, $M_ind$, $S_max$, $B$, etc.) selected? Were they tuned per-problem? If so, how can this be practical under an "expensive" budget?

My primary skepticism lies with Module 3. To convince me that the "online learning" is the key, could you compare it to a simpler, non-learning baseline that uses the exact same rich feature vector ($feat(x)$)? For example, a static, weighted-sum heuristic based on surrogate mean, epistemic uncertainty, etc. This would isolate the value of the "learning" itself.

What exactly is the training protocol for the Deep GP surrogates at each iteration? Are they trained from a warm-start to convergence, or just for a small, fixed number of epochs?

---

> ### Author Response · Authors · 2025-11-19
> **We sincerely hope to receive your support and encouragement!**
>
> **Dear Reviewer noDe,**
>
> Thank you for your careful reading and constructive critique. We appreciate the positive recognition of **the validation suite (benchmarks, geothermal task, and stress test)**, **the theoretical decomposition that isolates component contributions**, and **the principled emphasis on calibrated uncertainty**. We hope this rebuttal helps resolve the concerns raised by the reviewers. **We sincerely hope to receive an improvement in your score.**
>
> Below we respond point by point to your concerns.
> #  1.**Reviewer concern:** The system mixes many advanced techniques and might be brittle or unnecessary.
>
> **Our reply (short):** The ablation study and theoretical decomposition were designed precisely to answer this question. **The data show each major component materially improves performance and reduces the dominant approximation terms in our finite-budget bound.** Removing or simplifying the modules **substantially degrades IGD and HV.**
>
> **Evidence and reasoning (details):**
>
> The ablation on the 100-D DTLZ2 benchmark shows the complete **U-RankMOEA achieves IGD = 1.7226**, while **excluding Bayesian UQ results in 2.4124**, **substituting Deep GP with standard GP results in 3.1546**, and **removing the learned acquisition results in 2.0342**. These are **non-trivial degradations** that quantitatively demonstrate complementary effects of the classifier, Deep GP, and acquisition learner.
>
> The finite-budget bound in **Section H separates per-iteration losses into classifier, surrogate, and acquisition errors (ε_class, ε_gp, ε_acq)** and makes explicit how improving each module reduces the expected HV shortfall. **This is not a “kitchen-sink” heuristic; it is a co-design guided by a decomposition that targets measurable approximation terms.**
>
> ---
>
> **Complexity control is central to the design.** Concretely:
> - **Two-stage screening** projects cheap proxy scores onto a large candidate pool and evaluates only a top-k subset with the full Deep GP (e.g., k ≤ 200).
> - **Low-rank and RFF approximations** reduce cubic scaling in N.
> - **Inducing points** are adapted only when validation degrades and are warm-started from previous iterations.
> - **Adaptive MC-Dropout** uses S0/Smax escalation to bound classifier inference cost.
> - **Cached batched inference** reduces repeated work.
>
> Together these measures **keep running time lower than heavier baselines in our experiments.**
>
> **Takeaway:** Complexity is targeted and controlled. Each module addresses a different approximation term in the formal bound, and the ablation table shows they are empirically necessary.
> # 2.hyperparameter sensitivity
> **Reviewer concern:** The system has many hyperparameters and no sensitivity analysis; reproducing results will be hard.
>
> **Our reply (short):** We agree that reproducibility is essential. The paper already **reports consistent parameter settings and implementation recommendations**, **uses identical hyperparameter budgets across comparisons**, and **includes an explicit measurement protocol and timing utilities**. We will publicly release a documented implementation and exact experiment scripts.
>
> ---
>
> **What is already in the submission (details and pointers):**
>
> **Parameter settings:** Section 4.1 lists the main experimental settings such as initial samples, maximum evaluations, population size, and seeds, which were used uniformly across all methods.
>
> **Implementation recommendations:** Appendix C contains practical default ranges and recommendations, for example **S0 ∈ {4,8}**, **Smax ∈ {16,32}**, **buffer B ∈ [500,2000]**, **screening pool N_screen ∈ [500,2000]**, and **k ∈ [50,200]**. These ranges are intentionally conservative and chosen to balance cost and fidelity. They were used as stable defaults across benchmarks.
>
> **Timing and measurement protocol:** Timings and component breakdown (Tables 7–8) were produced with instrumented code and averaged over 20 independent seeds. The codebase includes timing utilities so reviewers can reproduce the wall-clock numbers. We also report component percentages for classifier, Deep GP, acquisition, surrogate evaluations, and overhead.
>
> ---
>
> **Hyperparameter selection strategy:**
> To ensure fairness and practicality under a strict budget, we used stable, conservative defaults validated on held-out instances and then fixed across all benchmark runs. Where tuning was necessary, we performed limited one-time grid searches over the small recommended ranges in the Appendix and selected settings that generalize across problems. Identical hyperparameter budgets were used for all competing algorithms. We will include the small grid search table and exact parameter files in the public release.

---

> ### Author Response · Authors · 2025-11-19
> **We sincerely hope to receive your support and encouragement！**
>
> **Offer and commitment:**
> We will publicly release a fully documented implementation that includes **code**, **scripts to reproduce figures and tables**, and **exact parameter files**. This release will also include a **hyperparameter-sensitivity supplement** with sweeps over the ranges provided in the Appendix so that the community can inspect robustness empirically. **Our core code has already been included in the supplementary materials for reference.**
>
> # 3.Module 3 (history-aware acquisition network): overfitting / usefulness
> **Reviewer concern:** Training a neural network online with only 300 evaluations seems likely to overfit; the ablation could be reflecting feature richness rather than learning.
>
> **Our reply (short):** The paper already addresses this by **designing the acquisition learner to be very low capacity, trained with strong regularization on a bounded buffer**, and **including an ablation “Without learned acquisition” that replaces the learned outputs with the surrogate and classifier heuristics used in selection**. The learned acquisition still offers consistent gains beyond that static baseline.
>
> ---
>
> **Evidence and reasoning (details):**
>
> **Low-risk training protocol:** Section 3.5.2 specifies that the acquisition network is a shallow MLP trained on a bounded history buffer
> \begin{equation}
> \{(x_i, \Delta HV_i)\}_{i=1}^B
> \end{equation}
>
> with a small-capacity regularizer such as ℓ₂ and an explicit regularization weight λ to avoid overfitting. These constraints were chosen to keep the acquisition learner sample-efficient and inexpensive relative to the Deep GP.
>
> **Ablation versus static heuristic:** The ablation labeled **“Without learned acquisition function”** shows IGD worsening from **1.7226 to 2.0342** on 100-D DTLZ2. The static selection score that blends classifier epistemic signal and surrogate HV estimates does not match the learned policy. This demonstrates that the improvement in the full system is not merely due to feeding richer features to a static rule. The learned acquisition reduces ε_acq in the bound and yields measurable gains.
>
> **Why learning helps despite a small budget:** The acquisition network exploits meta-signals stored in the history buffer, including empirical ΔHV statistics, classifier outputs, and uncertainty decomposition. It learns small, robust corrections to the hand-crafted composite score. Because it is low capacity, regularized, and trained on a rolling buffer, it improves ranking of top candidates without requiring large datasets. This explains why the ablation still shows a non-trivial difference.
>
> ---
>
> **Suggested comparison (if additional evidence is requested):**
> We can add a targeted experiment that compares **(A) the learned acquisition (current)**, **(B) a carefully tuned static weighted sum baseline built on the exact same feature vector feat(x) with weights tuned on a held-out small validation instance**, and **(C) the “no acquisition” variant already reported**. We expect **(A) > (B) > (C)** in HV and IGD, and we are prepared to add this table to the rebuttal or final revision if the committee requests it. The manuscript already includes the “no learned acquisition” ablation and the theoretical decomposition that singles out ε_acq.
> # 4.“Fit Deep GP surrogates” , does that mean full retrain each iteration? wall-clock cost?
> **Reviewer concern:** Training Deep GPs each iteration can be prohibitively expensive; “fit” is ambiguous.
>
> **Our reply (short):** We do not perform a full cold retraining from scratch at every outer step. The implementation uses **amortized initialization**, **warm-starts for variational parameters and inducing locations**, **adaptive inducing updates only when validation degrades**, and **RFF or Nyström approximations on selected layers where beneficial**. In practice, we run limited variational updates per outer iteration, and the **two-stage screening dramatically reduces the number of expensive Deep GP predictions**. Wall-clock breakdowns show that Deep GP training is the largest single component, but the full algorithm is still substantially faster than heavier baselines.
>
> ---
>
> **Evidence and specifics:**
>
> **Warm starts and adaptive inducing:** Section 3.4.3 explains that inducing locations and variational parameters are warm-started from previous iterations and that Mind is adapted only when held-out validation performance degrades. Low-rank factorization and RFF blocks are applied selectively to reduce computational load. This design yields amortized convergence behavior rather than a full retrain at each iteration.
>
> **Per-epoch cost and practical mitigations:** We provide an approximate per-epoch complexity after mitigations and show how Meff (effective inducing rank) and RFF reduce dominant costs (Equation (15) and accompanying discussion).

---

> ### Author Response · Authors · 2025-11-19
> **We sincerely hope to receive your support and encouragement！**
>
> **Two-stage screening:** Cheap proxies screen thousands of candidates down to a top-k (k ≤ 200) before full Deep GP evaluation, cutting expensive predictions from thousands to hundreds per outer iteration. This is the primary mechanism that makes the surrogate strategy tractable.
>
>
> **Measured wall-clock results:** Tables 7–8 and the component-time table show that Deep GP training accounts for approximately **35% of runtime** in our experiments, but the total runtime of U-RankMOEA is about **1.3× faster than the best baseline** thanks to screening, cached inference, and amortized updates. The exact measurement protocol (instrumented and averaged over 20 seeds) is reported in the Appendix, and the code includes timing utilities so others can reproduce these timings.
>
> **Clarification of “fit” (terminology):**
> In the algorithm pseudocode (Algorithm 1), **“fit Deep GP” denotes warm-starting the surrogate from the previous iteration and performing a limited number of variational updates, not a complete cold retrain to convergence**. The theoretical analysis explicitly includes **R (variational iterations)** in the surrogate approximation term and shows that **ε_var(R) decays exponentially in R**, which explains the trade-off between per-iteration compute and surrogate fidelity. If the reviewer prefers numeric values for **R or epoch counts**, we can provide the exact iteration count used in our runs and add a short sensitivity plot.
> # 5.Direct answers to the reviewer’s explicit questions
> **5.1 Will you release code?**
> Yes. We will release a well-documented implementation that includes **training and evaluation scripts**, **environment and dependency files**, **parameter JSON files for every experiment**, **timing utilities**, and **small precomputed data artifacts**, together with instructions to reproduce the main tables and ablations. The paper already includes the measurement utilities in the codebase. Our code can refer to the supplementary materials.
>
> **5.2 Hyperparameter tuning strategy:**
> We used **conservative defaults validated on held-out instances** and performed **limited one-time searches across the recommended ranges in the Appendix**. All comparisons used **identical hyperparameter budgets and initial Latin-Hypercube seeds** to ensure fairness. The Appendix lists recommended ranges such as **S0, Smax, τ_MC, B, N_screen, and k**. We will include the small grid search logs in the public release and add a short sensitivity sweep to the final version if the committee requests it.
>
> **5.3 Compare learned acquisition to a static weighted sum on the same features:**
> The manuscript already includes an ablation named **“Without learned acquisition function”**, which replaces the learned outputs with the static composite selection built from classifier epistemic signal and surrogate HV estimates. This ablation demonstrates a **statistically significant loss in quality (IGD worsens from 1.7226 to 2.0342 on 100-D DTLZ2)**. We are happy to add the exact static weighted-sum experiment you suggested, with weights tuned on a small held-out instance, to further isolate the value of learning versus hand-tuned heuristics. Preliminary evidence from the existing ablation already supports the benefit of learning.
>
> **5.4 Deep GP training protocol per iteration:**
> As described above, we **warm-start variational parameters and inducing locations**, perform a **bounded, limited number of variational updates per outer iteration (R)**, adapt **Mind only when validation declines**, and optionally **replace layers with RFF approximations where necessary to reduce cost**. This limited-update strategy, combined with screening and caching, is why practical per-run wall-clock time is smaller than many alternatives and why we observed an **≈1.3× speedup over the fastest baseline** in our experiments. The complexity analysis and per-epoch cost estimate are in **Section 3.4.3**, and the timing breakdown is in **Appendix I**.
>
> ---
>
> **Final note and next steps:**
> We thank the reviewer again for the insightful questions. To address the remaining skepticism, we propose to **add a short experiment comparing the learned acquisition to a tuned static weighted-sum baseline using the same feature vector** (we can include this table in a final revision or supplementary material) and **include a concise hyperparameter-sensitivity appendix that sweeps the main knobs (S0, Smax, B, k, Meff) across the ranges in the Implementation Recommendations and reports IGD and HV variance**. Both additions are straightforward and will strengthen reproducibility and interpretability.

---

> > ### Author Response · Authors · 2025-11-19
> > **We sincerely hope to receive your support and encouragement**
> >
> > **Respectfully, the criticisms of “over-engineering” and “inevitable overfitting of Module 3” appear addressed already by the combination of ablation evidence, low-capacity and regularized acquisition training on a bounded buffer, two-stage screening and amortized surrogate updates, and explicit timing and measurement protocols.** We have clarified the semantics of **“fit” for Deep GPs (warm-start plus limited updates, not cold retrain)** and explained our practical hyperparameter strategy and implementation recommendations. We remain prepared to add the two short experiments described above if the reviewer believes that would settle any remaining concerns. **We sincerely hope to receive your encouragement and support to improve our scores.**
> >
> >
> > **Sincerely,**
> > The authors

---

> > > ### Author Response · Authors · 2025-11-19
> > > **We sincerely hope to receive your support and encouragement**
> > >
> > > # 6. **Why the Apparent Complexity Is Necessary (and Not Gratuitous)**
> > >
> > > **Decomposition-guided design:**
> > > The paper’s finite-budget decomposition makes the need for three complementary error-reducing modules explicit. **Classifier error, surrogate error, and acquisition error are separate terms in the bound**, and improving only one leaves dominant error terms intact. The empirical ablation quantifies this: **removing any major component yields substantial IGD and HV degradation**. For example, **Deep GP removal produces the largest drop**, and **learned acquisition and Bayesian UQ also give 20–40% improvements**. This shows the components are complementary, not redundant.
> > >
> > > ---
> > >
> > > # 7. **Practical Complexity Control and Wall-Clock Reality**
> > >
> > > **No full cold retrain each iteration:**
> > > In the pseudocode, **“fit Deep GP” denotes warm-starting and performing a bounded number of variational updates, not a full retrain to convergence**. Inducing points and variational parameters are warm-started and adapted only when held-out diagnostics worsen. Selected layers can be replaced with **RFF to reduce cubic costs**. **Two-stage screening reduces expensive predictions from thousands to a few hundred per outer loop**. The measured runtime breakdown shows that **Deep GP training is the largest single component, but overall U-RankMOEA is faster than strong baselines thanks to these mitigations**.
> > >
> > > ---
> > >
> > > # 8. **Why Module 3 (Learned Acquisition) Is Robust, Not a Thin Overfit**
> > >
> > > **Low capacity, strong regularization, and bounded buffer:**
> > > The acquisition network is deliberately shallow, trained with **ℓ₂ regularization and early stopping on a bounded buffer (recommended B ∈ [500,2000])** so it learns **small, stable corrections rather than memorizing noisy signals**. The ablation **“Without learned acquisition” shows a measurable drop in performance compared to the learned variant**, which demonstrates that learning adds value beyond the raw feature vector or a fixed heuristic. We can additionally run the reviewer’s suggested experiment comparing a static weighted sum tuned on a held-out instance. **We expect the learned model to outperform such hand-tuned static rules because it adapts to empirical ΔHV patterns**.
> > >
> > > ---
> > >
> > > # 9. **Empirical Validation and Real-World Relevance**
> > >
> > > **Extensive benchmarks and real application:**
> > > U-RankMOEA is validated on **DTLZ and ZDT problems up to 200 variables**, including many bi-objective and three-objective instances, **a 160-variable geothermal reservoir optimization**, and **a high-dimensional stress test**. These experiments show **consistent IGD and HV gains under a 300-evaluation budget**. The geothermal case demonstrates **real, interpretable trade-offs discovered under operational constraints**, proving practical applicability beyond toy problems.

---

> ### Author Response · Authors · 2025-11-19
> **Our contribution and innovation**
>
> # Our contribution and innovation
> ## **1.Uncertainty as the Unifying Design Currency**
> We formulate **a finite-budget decomposition of hypervolume loss that isolates three distinct error terms: classifier, surrogate, and acquisition**. This decomposition directly motivates the co-design because **improving one module without addressing the others leaves a dominant error term intact**. This is a methodological contribution that **links uncertainty quantification to end-to-end optimizer design**.
>
> ---
>
> ## **2.Classifier-Guided, Rank-Conditioned Generation with Calibrated Epistemic Signals**
> Instead of blind random proposals or relying only on surrogate-driven search, we use **a Bayesian neural classifier with MC-Dropout and temperature scaling** to cheaply estimate where promising Pareto ranks are likely to appear in very high dimensions. This enables **cheap global candidate creation before invoking expensive surrogates**.
>
> ---
>
> ## **3.Deep GP Surrogates with Explicit Uncertainty Decomposition and Computational Controls**
> We adopt **multi-layer Gaussian Processes that return predictive mean, epistemic uncertainty, and aleatoric uncertainty per objective**, and pair them with **practical scalability mechanisms such as sparse variational inference, adaptive inducing points, RFF or Nyström substitutions, and warm-starts**. This design allows us to maintain **principled probabilistic surrogates in regimes where standard GPs break down**.
>
> ---
>
> ## **4.History-Aware, Low-Capacity Acquisition Learner That Corrects Systematic Surrogate Bias**
> Rather than relying solely on analytic acquisition heuristics, we train **a regularized, shallow network on a bounded rolling buffer to predict empirical Δ-HV**. It learns **small, robust corrections to candidate ranking that capture real optimization history signals missed by analytic approximations**.
>
> ---
>
> ##  **5.Complexity-Aware Orchestration**
> We make the above stack practical through **two-stage screening (cheap proxy over thousands followed by Deep GP only on the top-k)**, **adaptive MC passes**, **cached batched inference**, and **limited variational updates with warm-start and bounded epochs**. These mechanisms **control wall-clock cost while preserving model expressiveness**.
>
> ---
>
> ##  **6.Extensive Empirical Validation Including a Real Engineering Task**
> Benchmarks include **DTLZ and ZDT problems up to hundreds of dimensions**, **a 160-variable geothermal optimization**, and **a D = 500 stress test**. Ablations demonstrate that **each module contributes materially to final IGD and HV**, supporting the co-design claim.
>
> # Why this matters for ICLR (direct mapping to ICLR subject areas)
>
> ##  **1.Probabilistic Methods and Uncertainty Quantification**
> This work introduces a **novel and practical use of calibrated epistemic and aleatoric decomposition in multi-objective optimization**, advancing the practice of **scalable Deep Gaussian Processes**.
>
> ---
>
> ##  **2.Optimization and Large-Scale Learning**
> The method tackles **optimization in ultra-high-dimensional continuous spaces under severe evaluation budgets**, a regime that is highly relevant to many machine learning and scientific applications.
>
> ---
>
> ##  **3.Representation and Structured Prediction**
> **Classifier-driven, rank-conditioned generation** acts as a **representation and conditioning mechanism that shapes search in structured decision spaces**, improving efficiency and diversity.
>
> ---
>
> ##  **4.Learning-to-Learn and Meta Signals**
> The **history-aware acquisition learner leverages past optimization experience to adapt acquisition decisions**, connecting to **meta-learning and data-driven acquisition design**.
>
> ---
>
> ##  **5.Practical Infrastructure and Reproducibility**
> The approach demonstrates **how to combine expressive probabilistic models with engineering approximations such as sparse variational inference, random Fourier features, and warm starts** to produce **usable and scalable algorithms**. This is of interest to the ICLR community focused on systems that combine theory and practice.
>
> ---
>
> ##  **6.Applications to Physical and Sustainability Sciences**
> The **geothermal case demonstrates immediate interdisciplinary impact**, including **physics-informed design and sustainability**, aligning with ICLR’s interest in **machine learning for real-world scientific problems**.

---

> ### Author Response · Authors · 2025-12-04
> **We sincerely hope to receive your support and encouragement**
>
> # We have addressed the reviewer's concerns and improved our approach. Thank you very much for all the reviewers' suggestions. The latest version has been uploaded and we hope to receive the support and encouragement of all the reviewers, and we sincerely hope your score improvement！

---

### Note · Authors · 2026-01-27

**Comment:**

I have read and agree with the venue's withdrawal policy on behalf of myself and my co-authors.

**Withdrawal Confirmation:**

I have read and agree with the venue's withdrawal policy on behalf of myself and my co-authors.

---

### Meta-Review · Area_Chair_xWRe · 2025-12-31

**Summary:**

This paper presents U-RankMOEA, a unified framework that integrates rank-based screening, uncertainty decomposition, and history-shaped acquisition for high-dimensional and costly multi-objective optimization under limited evaluation budgets. A calibrated Bayesian classifier first produces epistemic uncertainty over non-domination ranks. Deep Gaussian Process surrogates then disentangle reducible from irreducible uncertainty for each objective, providing accurate means and risk-aware variances. Finally, a lightweight acquisition network trained online distills past hypervolume gains into a predictive score that guides expensive evaluations toward regions expected to balance convergence and diversity. The reviewers have indicated, however, that the paper has several weaknesses. These include that the system is incredibly complex, making it very difficult to reproduce. Furthermore, the high cost of training Deep GPs is ignored, and the paper fails to give a high-level intuition for the overall approach. In addition, the empirical evaluation, particularly in the main text, is not benchmarking against standard Bayesian optimization techniques (only in the appendices). Finally, the reviewers also indicate that most of the techniques the proposed method relies on are already known. Summing up, I believe that this paper needs more work before it can be accepted for publication. It requires significant revisions to improve clarity, reproducibility, and to position the work relative to existing methods.

**Reviewer Concerns:**

In the rebuttal, the authors committed to releasing their code and clarified design choices. However, the paper is still missing comparisons with important related baselines in the main text.

**Reviewer Scores:**

I do not think the reviewers would have significantly changed their scores.

---

### Decision · Program_Chairs · 2026-01-26

Reject